# High Frequency of Non-Compliance with Quality Indicators of Enteral and Parenteral Nutritional Therapy in Hospitalized Patients

**DOI:** 10.3390/nu12082408

**Published:** 2020-08-12

**Authors:** Daiane Aparecida Nogueira, Lara Princia Ferreira, Renata Paniago Andrade de Lúcia, Geórgia das Graças Pena

**Affiliations:** 1School of Medicine, Federal University of Uberlândia, Uberlândia-MG 38.405-320, Brazil; daiane_nogueira14@hotmail.com (D.A.N.); laaraprinciaferreira@hotmail.com (L.P.F.); 2Management of Nutrition and Dietetics, Clinical Hospital of the Federal University of Uberlândia, Uberlândia-MG 38.405-320, Brazil; renatap@hc.ufu.br

**Keywords:** quality indicators, enteral nutrition, parenteral nutrition, nutritional therapy, health care, quality management

## Abstract

Quality indicators in nutritional therapy (QINT) are measures of the effectiveness and quality of nutrition support. The purpose of this study was to evaluate the frequency of the QINT adequacy of Enteral Nutritional Therapy (EN) and/or Parenteral (PN) in hospitalized patients and identify the best indicators according to health professionals. A prospective study was performed, including data from patients aged 18 years or over admitted to clinical or surgical wards. The patients who had received EN and/or PN were followed from the first day of nutritional prescription until discharge. Twelve indicators were calculated, as recommended by the literature. Regarding professional opinion, the QINT adequacy was evaluated by observing its utility, simplicity, objectivity, and cost. Of the 727 hospitalized patients, 101 were on EN and/or PN. Regarding the 12 QINT evaluated, only 25% (3) achieved the goals: involuntary withdrawal of enteral feeding tube (0.01%); feeding tube occlusion or withdrawal per occlusion (0%); the measurement of energy and protein requirements (92%). A high frequency of non-compliance (75% of QINT) was observed in clinical and surgical patients on EN and/or PN. With knowledge of the six best indicators chosen by health professionals in this service, it will be possible to elaborate protocols according to the real-life situation in the institution.

## 1. Introduction

Improvement of healthcare quality must be a priority to ensure patient care as best as possible. Studies about this topic are scarce, although more than 9% of hospitalized patients are harmed by adverse events [1] EN, resulting in malnutrition or clinical complications such as increased morbidity and mortality, increased length of hospitalization, more frequent re-admission, and increased healthcare costs [2,3,4,5]. Nevertheless, patients frequently do not receive evidence-based treatments [6].

This scenario needs changes regarding quality improvement interventions aimed at developing health team awareness and enhancing their decision-making skills to develop more adequate and efficient interventions, resulting in improved care and patient outcomes [7].

Nearly half of hospitalized patients have some degree of malnutrition and require nutritional therapy [2]. Although there is a high prevalence of malnutrition, there are few studies considering quality indicators in nutritional therapy (QINT) [8,9,10,11,12,13]. This is important, as these indicators minimize energy and protein imbalance and weight loss, and also help prevent adverse outcomes and clinical complications, including increased infection rate, impaired wound healing, a longer length of hospital stay, higher treatment costs, and mortality [14]. However, the prevention and minimization of these complications requires careful monitoring and the skills of multidisciplinary nutrition therapy teams (MNTT), aiming at quality nutrition care [15].

The task force of clinical nutrition of the International Life Sciences Institute (ILSI), aiming at better quality control, proposed a list of 36 quality QINT validated by 41 experts [16]. In 2012, the 10 most important indicators were selected [7]. These facilitate the accurate assessment of nutritional care in healthcare services [1,16].

Most of the studies about QINT investigated critically ill patients [8,9,10], mainly because these patients had intensive monitoring and follow-up due to the high risk of clinical and infectious complications and also had high mortality rates. We found only three studies involving quality indicators with clinical [11], surgical [12], or mixed populations [13]. Regarding clinical and surgical patients, approximately 61% were malnourished [3] and usually presented some nutritional deficit, mainly due to trauma response, insufficient food intake, or nutrient malabsorption when associated with gastrointestinal diseases [2]. Despite these statistics, only two studies investigated patients by their clinical record data [8,12], and these either did not evaluate all the top 10 indicators as suggested by the ILSI in full [16] or just evaluated enteral nutritional therapy (EN) patients [8]. To the best of our knowledge, this is the first study with clinical and surgical patients in EN and parenteral nutritional therapy (PN) to evaluate prospectively all top 10 indicators, as suggested by ILSI-Brazil.

Lastly, health quality indicators should be practical, useful, and considered important by the health team, so the professionals’ opinion is important to establish the indicators that are appropriate to the day-to-day reality of the service. Health professionals should define which QINTs are needed and applicable in services in order to improve and prioritize the nutritional care for hospitalized patients. Therefore, the aim of this study was to evaluate the frequency of QINT adequacy in clinical and surgical patients during EN or PN and to identify those indicators that were, in the opinion of the health professionals, the most applicable for a large health service.

## 2. Materials and Methods

### 2.1. Study Design

A prospective study was conducted from November 2017 to May 2018 in a large reference tertiary hospital. All the hospitalized patients in clinical and surgical wards aged 18 years or older and who had been on EN and/or PN for ≥24 h during hospitalization were included. Patients who were not hospitalized in clinical and surgical wards or who were aged under 18 years old were excluded.

In order to achieve the second objective of this study, the professional dietitians, nurses, physicians, and pharmacists involved with nutrition support (EN and PN) were invited to select the indicators they believed to be most useful for the specific service. The Research Ethics Committee approved this study (CAAE: 79696317.7.0000.5152).

### 2.2. Data Collection

Daily clinical and nutritional data were collected from clinical records, considering the first day of admission to the ward until the outcome (ward/hospital discharge or death). Gender, age, admission date, length of hospitalization, medical specialty, subjective global assessment (SGA) [17], nutritional reassessment, energy and protein requirements, characteristics of the enteral diet, gastrointestinal complications, nutritional therapy indication criteria collected from their clinical records, and fasting time were registered. Thus, all the data on nutritional therapy in the follow-up were obtained. In order to determine a viable number of indicators applicable to clinical practice in the context of this specific hospital, the QINT were evaluated according to the top 10 developed by ILSI-Brazil [16] and selected by Verotti and colleagues in 2012 [7].

SGA was used to estimate the frequency of carrying out nutrition screening within 24 h of admission and also to estimate the application of the nutrition assessment indicator, because it is a nutrition diagnosis method. Hyperglycemia and hypoglycemia were defined according to values of >160 mg/dL [16] and <70 mg/dL [18], respectively. According to the hospital protocol, diarrhea was defined as three or more liquid bowel movements a day [19]. To estimate the other QINT, all the data registered in clinical records and forms by the multi-professional team were considered, as shown in Appendix A.

Considering the demands of the hospital and aspects related to nutritional care, we added two indicators (the nutritional reassessment of patients on EN and the adequacy of the prescribed EN volume versus the administered) that we considered viable and useful to estimate in this health service. Besides that, the reassessment and the balance of the diet are important for the nutritional follow-up, providing a guarantee of energetic-protein contribution to the patient [20,21]. Thus, 12 QINT were estimated in patients on EN and/or PN. In order to classify the adequacy of each QINT, we used the Task Force of Clinical Nutrition of ILSI-Brazil criteria [16] and other literature recommendations [22]. The formulas and goals are also shown in Appendix A.

### 2.3. Professional Opinion and Selection of the Quality Indicators for Nutrition Therapy

Psychometric scales and statistics were used to analyze the reliability of a set of 12 QINT according to professional opinion, following the method published by Verotti and colleagues in 2012 [7]. We invited 60 professionals with experience and/or specialization in nutritional therapy from the clinical and surgical wards to complete the questionnaire, and 49 professionals accepted.

The opinion of these professionals was obtained through a questionnaire applied in a face to face interview. First, the attributes were explained: utility—“the QINT should be useful, advantageous, and valid?”; simplicity—“is this QINT simple to search, calculate, and analyze? If so, the greater the chances and opportunities for use”; objectivity—“has this QINT a clear goal, increasing the reliability of what is pursued?”; low cost—“will the cost of doing this QINT limit its routine use?” [23].

Professionals were asked to score the attributes following a 5-point Likert scale (0 = very bad, 1 = bad, 2 = indifferent, 3 = good, and 4 = very good) [7,24]. The QINT was identified from the top 5 scores (arithmetical average of the 4 assessed attributes for each QINT) obtained by the dependency of the adequate reliability.

### 2.4. Statistical Analysis

Statistical Package for Social Sciences software (SPSS, version 17.0, Chicago, IL, USA) was used to estimate the descriptive statistical and Cronbach’s alpha, and each QINT was evaluated as follows: excellent > 0.9; good > 0.8 to ≤0.9; acceptable > 0.7 to ≤0.8; questionable > 0.6 to ≤0.7; poor > 0.5 to ≤0.6; unacceptable ≤ 0.5 [25]. Data were expressed as the absolute number and percentage.

## 3. Results

During the period, of the total 727 hospitalized patients, 101 were given nutritional therapy (main sample), 78 patients were on EN, and 23 patients were on PN (Figure 1). Most of them, 76.2%, were in surgical wards, and 23.8% were in clinical wards. More than half were male, with the mean age being 57.8 ± 16.1 years (Table 1).

According to SGA, 55.4% of patients had some level of malnutrition and 31.7% did not have a reported nutritional diagnosis (Table 1). The main criteria for professional dietitians and physicians to indicate EN were “lowering level consciousness” (22.1%) and “insufficient dietary intake” (22.1%) (Table 1). The mean length of stay was 20.36 ± 15.95 days. The patients were maintained on EN or PN for 17.41 ± 14.22 and 13.62 ± 7.99 days, respectively. The main reason for fasting for ≥24 h during the EN administration was scheduled examinations 40% (20), such as colonoscopy, gastrostomy, jejunostomy, endoscopy, X-ray, magnetic resonance, and catheterization, and 14% of the clinical records did not show the reason (Figure 2).

After applying the 12 QINT, 75% (9) of those investigated were found to be non-compliant, according to the proposed goals. Compliance was only found with three QINT: the involuntary withdrawal of the enteral feeding tubes, the occlusion of the feeding tubes, and the measurement of energy expenditure and protein needs in patients on nutritional therapy (Figure 3—the light bars indicate the recommended goals and the dark bars indicate the results obtained in the present study). It is important to mention that the first five indicators are extremely important for the good clinical and nutritional monitoring of patients, so we hope to reach and/or exceed the recommended goal. The last seven indicators, on the other hand, are adverse events in the clinical management of patients that must be avoided, so that the objective is to achieve values below the goals.

Regarding professional opinion, 49 professionals involved in EN and PN in the hospital were interviewed (dietitians, *n* = 22; nurses, *n* = 21; physicians, *n* = 3; pharmacists, *n* = 3). Of the 22 dietitians and 3 invited pharmacists, all accepted and answered the questionnaire. In addition, 25 nurses accepted the invitation, but only 21 nurses answered the questionnaire. Of the 10 physicians invited, 1 physician declined the invitation and 6 physicians accepted but did not answer the questionnaire, resulting in a total of 3 physicians.

According to the professionals’ opinions, the six best QINT were digestive fasting for more than 24 h, glycemic dysfunction on EN/PN, carrying out nutrition screening, tube feeding occlusion, the involuntary withdrawal of enteral feeding tubes, and diarrhea in patients on EN. Table 2 provides a description of the QINT indicators, considering the average scores according to the opinions of the health professionals.

## 4. Discussion

This study showed a high frequency (75%) of non-compliance with QINT in clinical and surgical patients on EN and PN regarding the evaluation of screening or nutritional diagnosis, feeding route, or follow-up complications. The six most important QINT for the service, according to the health professionals, were related to follow-up complications and screening or nutritional diagnosis. Most of them were found to be non-compliant in the evaluated service.

In the present study, the SGA was used for the first and second QINT, since that is recommended for nutritional risk as well as nutrition status assessment. Detecting nutritional risk by performing nutritional screening allows the nutrition support team to take early initiative, even in patients with apparently adequate total body weight, whereas SGA is primarily effective in recognizing current malnutrition [7,26,27,28,29]. Many studies have demonstrated a deficiency in screening nutrition within 24 h of admission, assessment, and even nutritional reassessment [8,9,10,13]. In studies involving clinical and surgical patients, the nutritional assessment corresponded to 30.7% and 75%, respectively [11,12], which agrees with the current study, the nutritional assessment being achieved in 55.5% of cases. When nutritional assessment is performed, there is possibly an early identification of malnutrition, which is an independent risk factor impacting on higher complications and increased mortality, length of hospital stay, and costs [30]. Thus, in order to obtain the required improvement in the quality of healthcare, adequate monitoring in these patients is needed to minimize the negative impacts and generate better clinical results.

After performing the nutritional assessment and diagnosis, nutritional requirements and a feeding route are necessary, as is the recording of daily follow-ups. There is a large gap in the literature about the criteria and fulfillment of these steps to indicate the enteral nutrition route and assess the adequacy of dietary volume. Few studies show the adequacy of the enteral diet volume by QINT [31,32], but these accord with the present study shows that patients usually receive a smaller enteral diet volume than what was prescribed. The study of Cartolano et al [8] shows that the adequacy of energy and protein supply minimizes infectious complications and contributes to a better prognosis. Besides that, when the enteral diet balance is less than 70%, it impacts significant associations between increased mortality and length of hospital stay [33]. The main reasons for non-administration (66.7%) in the present study were fasting for exams and surgery, patient refusal due to the feeling of gastric fullness, the repositioning of the tube, and gastrointestinal complications (vomiting, bloating, abdominal distension). In this sense, it is very important to perform the daily monitoring of nutritional supply, to identify possible causes of interference, and to establish strategies and protocols to guarantee the best possible care for patients.

Besides the monitoring of the enteral diet balance, it is also very important to monitor and control nutrition therapy-related clinical variables such as fasting time, glycemic dysfunction, central venous catheter infection, and diarrhea. Studies [2,3,34] show the impact of fasting, which should be monitored to prevent glycemic alteration and to prevent malnutrition, the high prevalence of which leads to greater susceptibility to infection. The American Society for Parenteral and Enteral Nutrition (ASPEN) [21] guideline recommends that diarrhea should not be a reason for the automatic discontinuation of enteral nutrition. Another one noted that the most severe clinical complication of PN is catheter-related infection, but there are not enough studies to point out PN as an isolated cause of infection [35,36].

Scientific meetings held between the MNTT, anesthesia and surgery teams can contribute to the control of complications related to nutritional therapy, since interventions to improve quality are often designed to enhance the therapies and use the data routinely collected in clinical practice. The main objective of these meetings is to determine the effect of an intervention based on the behavioral change of health professionals, typically manifested as adherence to an optimal health care process [6].

Given the above, there is a need for the constant assessment of the quality of nutritional therapy aimed at correcting non-conformities and ensuring effective nutritional support to patients who need it. This assessment should be performed using quality indicators. The Joint Commission on Accreditation of Health Care Organization recognized over a decade ago the need for the constant evaluation and monitoring of patients who are given nutrition therapy [37]. Thus, the QINT are an appropriate tool to better evaluate the quality of nutritional assistance provided in health services [1]. Currently, the definition of which QINT are practical in clinical nutrition services is a challenge to nutritional therapy experts. There are no standardized rules to establish quality indicators, as these become clear as the consequences of the needs and experiences developed at each health institution [10].

Regarding professional opinion, we observed non-compliance in the majority of the six QINT selected by the health professionals as the most important for the service and for which evaluation was needed. The dietitians and nurses interviewed agreed with these six selected QINT, except for involuntary tube withdrawal and the occlusion of tube feeding. In the medical category, the Cronbach’s alpha values did not show consistency. When we analyzed the opinion of the pharmacists, we found that only two of their chosen QINT were on the list of six (glycemic dysfunction and carrying out nutrition screening).

The two studies that selected QINT by the Likert scale method were conducted specifically with nutritional therapy professional specialists [7,24]. In the present study, 14 dietitians were specialists, whereas among the physicians, nurses, and pharmacists, only one of each was a specialist. Often, the lack of attention to clinical nutrition during undergraduate medical, nursing, and pharmacy education results in professionals who are not fully aware of the importance of nutrition [20]. It is necessary to have more specialized professionals involved in nutritional therapy, favoring the better management of nutritional quality in the service. Therefore, the differences in the QINT ranking among health professionals probably occur due to the different areas of expertise, in which some themes are more valorized them others during the undergraduate programs. In this sense, the specializations courses and clinical experiences before could modify the valorization about some clinical signals during the management of complications.

Thus, MNTTs are extremely important in order to establish management and control criteria for EN and PN, as well as having specialists with different skills [10]. However, MNTTs and protocols alone do not guarantee the practice of QINT. The continuous training of healthcare professionals and routine audits are necessary to improve the following of institutional protocols.

The present study has some limitations. The clinical records are completed by the professionals and could have some data loss. Although we were unable to ask the professionals directly about the data, we read all the clinical records and data sheets completed by all the professionals for every day of the study. Regarding the selection of six QINT, most of the interviewed professionals had no specialization in nutritional therapy or experience with the Likert scale, which could compromise their scoring of the attributes, but this bias was reduced because the interviewed professionals worked directly with patients on nutritional therapy. A limited number of pharmacist and physician professionals were experienced in nutritional therapy and aware of the importance of this type of therapy, and this was reflected in the low number of returned questionnaires. It is noteworthy that the study is pioneering the introduction of quality concepts was based in a public university tertiary hospital. In our study, the six QINT were chosen based on the opinion of professionals in the analyzed institution and may not be suitable for universal use.

## 5. Conclusions

Therefore, a high frequency of non-compliance (75% of QINT) was observed in clinical and surgical patients during EN and PN. This revealed the lack of nutritional assessment and the need to improve nutritional therapy monitoring, given the discrepancy between the nutritional requirements and the real-life situation. The professionals’ opinions involved indicators related to the follow-up of nutritional therapy, non-compliant in the evaluated service. We suggest that the evaluated health service consider the six best indicators found by the present study. We should note that the indicators were evaluated based on the opinions of the professionals of the institution, and may not be suitable for universal use. Therefore, it would be advisable that each service should develop its own QINT contemplating their own goals. The elaboration, surveillance of protocol application, and training and awareness of professionals about the importance of QINT are essential.

## Figures and Tables

**Figure 1 nutrients-12-02408-f001:**
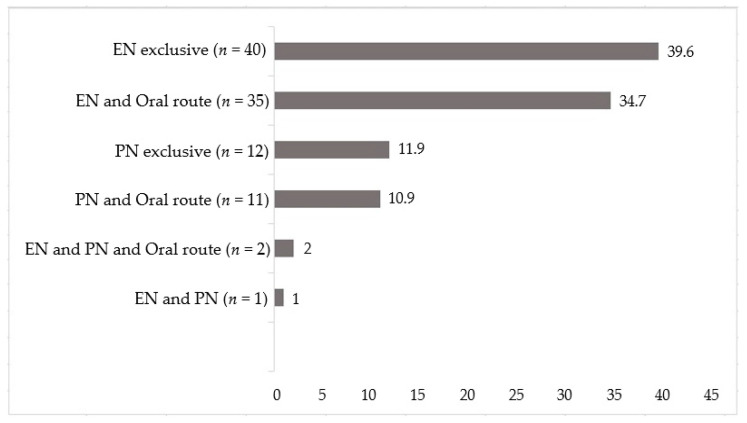
Nutritional support of surgical and clinical patients using Enteral Nutrition (EN), Parenteral Nutrition (PN), or the oral route (%), *n* = 101.

**Figure 2 nutrients-12-02408-f002:**
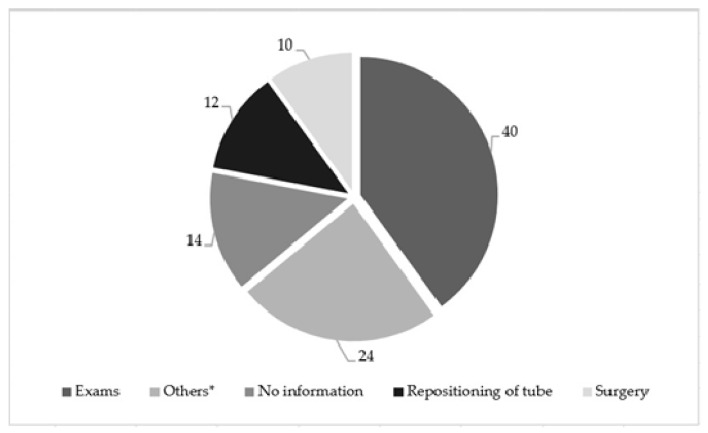
Factors leading to fasting for ≥24 h (%) during enteral nutritional therapy administration, *N* = 101. * Phonoaudiologist assessment; lowering level of consciousness; dose noradrenaline; hemodynamic instability; drainage of feeding tube; methylene blue test.

**Figure 3 nutrients-12-02408-f003:**
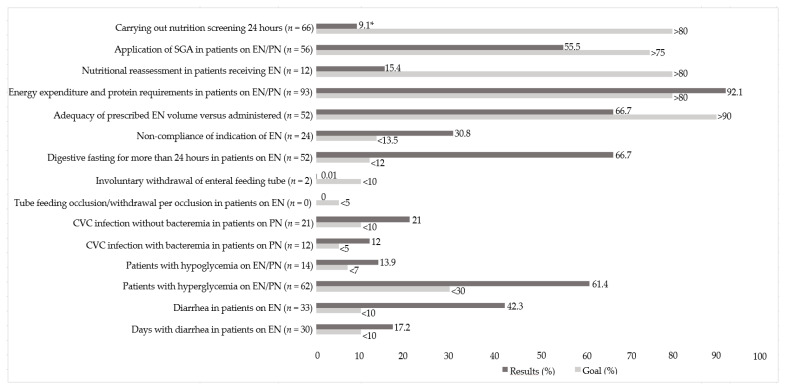
Quality indicators in enteral and parenteral nutrition therapy and goals (%). (* *N* = 727; *N* = 101; ENT = 78; PNT= 23) Abbreviation: SGA, subjective global assessment; EN, enteral nutrition; PN, parenteral nutrition, CVC, central venous catheter. Light bars indicate the recommended goals and the dark bars indicate the results obtained in the study.

**Table 1 nutrients-12-02408-t001:** Demographic, clinical, and nutritional characterization of data from the clinical and surgical patients using enteral and parenteral nutritional therapy (*N* = 101).

Variables	*N*	%
Sex		
Male	58	57.4
Female	43	42.6
Age group (years)		
<60	54	53.5
≥60	47	46.5
Subjective Global Assessment		
Well nourished	13	12.9
Moderate malnutrition	47	46.5
Severe malnutrition	9	8.9
No information	32	31.7
Patient Profile—Ward		
Surgical	77	76.2
Clinical	24	23.8
Medical specialty		
General surgery	39	38.6
Neurology/Neurosurgery	14	13.8
Other *	12	12
Internal medicine	9	8.9
Digestive surgery	8	7.9
Surgery oncology	7	6.9
Orthopedics/Traumatology	5	4.9
Urology	4	4
Gastroenterology	3	3
Indication criteria Nutrition Therapy Enteral **		
Lowering level consciousness	17	22.1
Insufficient food intake/<60% Total caloric value	17	22.1
Improvement in caloric intake	13	16.9
No information	8	10.4
Dysphagia	7	9
Non-functioning gastrointestinal tract	4	5.2
Nausea, vomiting, gagging	4	5.2
Other ***	4	5.2
Malnutrition	3	3.9
Characteristic Enteral Nutrition Therapy		
Polymeric Normocaloric and Hyperproteic (1.2 kcal/mL)	32	41.6
Polymeric Hypercaloric and Hyperproteic (1.5 kcal/mL)	25	32.5
Oligomeric (1.0–1.2 kcal/mL)	15	19.5
Polymeric Normocaloric and Normoproteic (1.0–2.0 kcal/mL)	4	5.1
Polymeric Specialized (Dialytic Renal) (2.0 kcal/mL)	1	1.3
Gastrointestinal Complications		
Diarrhea	33	42.3
Constipation	31	30.7
Abdominal distension	28	27.7
Abdominal pain	26	27.7
Vomiting	9	8.9
Melena	2	1.9
Outcome		
Discharge hospital	87	86.1
Death	14	13.9

* Cardiology; head and neck surgery; vascular surgery; nephrology; oncology; otolaryngology; proctology; heumatology. ** Criteria as reported in clinical records. *** Megaesophagus; risk bronchoaspiration; transition from diets; oral trauma.

**Table 2 nutrients-12-02408-t002:** Classification of 12 quality indicators in enteral and parenteral nutritional therapy according to opinions of health professionals.

	Rank—All (*n* = 49)	Mean	α *	Rank—Dietitians (*n* = 22)	Mean	α *	Rank—Nurses (*n* = 21)	Mean	α *	Rank—Physicians (*n* = 3)	Mean	α *	Rank—Pharmacist (*n* = 3)	Mean	α *
Frequency of digestive fasting for more than 24 h in patients on oral nutrition or EN	1	14.22	0.857	1	15.45	0.729	1	13.00	0.872	7	14.66	0	9	13.33	1
Frequency of patients with glycemic dysfunction on EN and PN	2	13.97	0.751	4	14.63	0.667	2	12.85	0.687	4	15.00	0	1	16.00	0.716
Frequency of carrying out nutrition screening of hospitalized patients	3	13.77	0.640	3	14.68	0.557	6	12.52	0.673	5	14.66	0	5	15.00	0
Frequency of tube feeding occlusion in patients on EN	4	13.71	0.911	2	15.04	0.851	7	12.14	0.914	2	15.33	0	8	13.33	1
Frequency of involuntary withdrawal of enteral feeding tubes	5	13.67	0.861	8	14.59	0.684	5	12.52	0.888	1	15.66	0	10	13.00	0.982
Frequency of diarrhea in patients on EN	6	13.67	0.754	5	14.63	0.562	4	12.57	0.788	6	14.66	0	7	13.33	1
Frequency of adequacy of prescribed EN volume versus administered	7	13.57	0.646	6	14.63	0.553	3	12.71	0.665	9	14.33	0	12	11.00	0.642
Frequency of CVC infection in patients on PN	8	13.30	0.819	9	14.36	0.678	8	11.71	0.823	3	15.33	0	6	14.66	1
Frequency of indication compliance of NT	9	13.08	0.852	7	14.61	0.827	10	11.14	0.844	8	14.66	0	4	15.00	1
Frequency of nutritional reassessment in patients receiving EN and ONS	10	12.91	0.794	10	14.31	0.603	9	11.57	0.705	11	12.33	0.970	11	12.66	0.923
Frequency of application of SGA in patients on EN and PN	11	12.37	0.811	11	14.27	0.256	12	10.09	0.845	12	12.00	0.667	2	16.00	0
Frequency of measurement or estimation of energy expenditure and protein needs in patients on NT	12	12.37	0.701	12	13.59	0.564	11	10.80	0.693	10	12.66	0.811	3	15.00	1

* α: Cronbach’s alpha. Abbreviations: EN, enteral nutrition; PN, parenteral nutrition; ONS, oral nutritional supplementation; NT, nutritional therapy; SGA, subjective global assessment; CVC, central venous catheter.

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
