# Peer review of "High Frequency of Non-Compliance with Quality Indicators of Enteral and Parenteral Nutritional Therapy in Hospitalized Patients"

_nutrients, 2020, doi:10.3390/nu12082408_

Round 1

Reviewer 1 Report

this study is overall very interesting. I found the introduction, methods, results, and discussion over QINT insightful.

I was surprised that so few patients were screen within 24hrs for malnutrition. I also found extremely interesting the different view physicians compared to allied healthcare professionals attribute to the ranking of these QINTS. For example, CVC bacteremia is quite high for physicians.

Comments:

In the discussion, could you please discuss more the potential explanations behind the difference in ranking, the authors found?

Also, Figure 3 I think needs some more explaining. It took some time for me to understand that the lighter color was the desired goal which wasn't achieved in most cases. This simply needs to be pointed out in the manuscript. 

Author Response

Uberlândia, Brazil

July 18th, 2020

Dear Reviewer#1,

Nutrients

Thank you very much for your work in revising our manuscript nutrients-863805 entitledHigh frequency of non-compliance with quality indicators of enteral and parenteral nutritional therapy in hospitalized patients. I am very pleased to receive the reviewer comments of the manuscript as "minor revision".

We sincerely appreciate all valuable comments and suggestions, which helped us to improve the manuscript. Our responses to the Reviewers’ comments are described below in a point-to-point manner. We hope that the reviewers will find our responses to their comments satisfactory.

Sincerely,

Geórgia das Graças Pena

Graduate Program in Health Sciences

Federal University of Uberlandia

Uberlandia, Minas Gerais, Brazil

ANSWERS TO THE REVIEWER’S COMMENTS

(The responses are shown immediately below the reviewer’s comments. The author’s responses are in red).

Reviewer #1:

This study is overall very interesting. I found the introduction, methods, results, and discussion over QINT insightful.

Comment to Reviewer#1

We appreciate the reviewer's comment and the opportunity to improve the manuscript.

I was surprised that so few patients were screen within 24hrs for malnutrition. I also found extremely interesting the different view physicians compared to allied healthcare professionals attribute to the ranking of these QINTS. For example, CVC bacteremia is quite high for physicians.

Comment to Reviewer#1

We appreciate the reviewer's comment and were also surprised by the low percentage of patients who received nutritional screening in 24 hours. We would like to emphasize that all the data were carefully collected and revised. So, this result indicates the urgent need for better management of nutritional quality in the evaluated service. Since the University Hospital evaluated is an important reference of the public health service, this manuscript alerts about this situation and will help in change this context in the next future.

Comments:

In the discussion, could you please discuss more the potential explanations behind the difference in ranking, the authors found?

Comment to Reviewer#1

We appreciate the reviewer’s comment. Probably, others potential explanations behind the differences in the QINT ranking among professionals are: i) Expertise of area – During the undergraduate courses, there is a different focus, or valorization for each to the expertise of the area. It is common, for example, medical students did not learn so much about Nutrition subjects and, consequently did not valorize the negative consequences of fasting or malnutrition, as much as a dietitian. ii) The different clinical experiences and points of view on the management of clinical complications. For example, an experienced nurse could understand better the negative impact of malnutrition in a surgery process because he saw negative outcomes many times in malnourished patients, while a young nurse could not achieve this. So, the first one is capable to observe and valorize the most important signals clinical when compared with the second one. In order to clarify these points, please, see the new sentences in the Discussion section (Page 10, lines 243-249).

Also, Figure 3 I think needs some more explaining. It took some time for me to understand that the lighter color was the desired goal which wasn't achieved in most cases. This simply needs to be pointed out in the manuscript.

Comment to Reviewer#1

We appreciate and agree with the Reviewer. We try to clarify this point in the text, where the light bars are the goals and the dark bars are the results obtained in the study. For a better understanding, please see the new sentences in the Results sections (Page 5, Lines 153-158) and Figure 3 (page 6).

            "…(Figure 3 -– the light bars indicate the recommended goals and dark bars indicate the results obtained in the present study.) It is important mentioning that the first five indicators are extremely important for the good clinical and nutritional monitoring of patients, so we hope to reach and/or exceed the recommended goal. The last seven indicators, on the other hand, are adverse events in the clinical management of patients that must be avoided, so that the objective is to achieve values below the goals. "

Reviewer 2 Report

The authors highlight the frequency of non-compliance with quality indicators of enteral and parenteral nutrition therapy in hospitalized patients. The authors carried a prospective study to assess compliance with 12 key indicators (10 from the literature and 2 proposed by the authors) and also surveyed specialists about the top 6 indicators for quality of nutrition therapy. I have given my suggestions below:

  1. Please specify exclusion criteria for the prospective study.
  2. Of 727 patients, only 101 were given nutritional therapy - this seems very odd for a non-ICU cohort. Please explain why this is so and why so many did not get nutrition therapy.
  3. Figure 3 is a bit confusing to comprehend - would suggest clearly spelling out the aim of each of the two bars in each category. In some categories, we want the indicator to be less and in other, we want it to be more - this does not come across very easily until I read it a few times.
  4. Many of the references are non-English and may need either English translations or citations to English language articles based on journal policy.

Author Response

Uberlândia, Brazil

July 18th, 2020

Dear Reviewer#2,

Nutrients

Thank you very much for your work in revising our manuscript nutrients-863805 entitledHigh frequency of non-compliance with quality indicators of enteral and parenteral nutritional therapy in hospitalized patients. I am very pleased to receive the reviewer comments of the manuscript as "minor revision".

We sincerely appreciate all valuable comments and suggestions, which helped us to improve the manuscript. Our responses to the Reviewers’ comments are described below in a point-to-point manner. We hope that the reviewers will find our responses to their comments satisfactory.

Sincerely,

Geórgia das Graças Pena

Graduate Program in Health Sciences

Federal University of Uberlandia

Uberlandia, Minas Gerais, Brazil

ANSWERS TO THE REVIEWER’S COMMENTS

(The responses are shown immediately below the reviewer’s comments. The author’s responses are in red).

Reviewer #2:

The authors highlight the frequency of non-compliance with quality indicators of enteral and parenteral nutrition therapy in hospitalized patients. The authors carried a prospective study to assess compliance with 12 key indicators (10 from the literature and 2 proposed by the authors) and also surveyed specialists about the top 6 indicators for quality of nutrition therapy. I have given my suggestions below:

Comment to Reviewer#2

We appreciate the reviewer's comment and the opportunity to improve the manuscript.

  1. Please specify exclusion criteria for the prospective study.

Comment to Reviewer#2

We understood the reviewer's concern. The exclusion criteria were i) hospitalized patients who were not in clinical and surgical wards,  and ii) patients aged under 18 years old. We had only 2 patients excluded by age criteria. There was not found any patients who were in enteral nutrition and/or parenteral nutrition ≤ 24 hours during hospitalization. In order to clarify these points, we add new sentences in the manuscript.  Please, see the new Methods (Page 2, lines 74-75).

  1. Of 727 patients, only 101 were given nutritional therapy-this seems very odd for a non-ICU cohort. Please explain why this is so and why so many did not get nutrition therapy.

Comment to Reviewer#2

We appreciate the reviewer comment. We revise all collect data carefully. The results are correct. Since the study was conducted with patients hospitalized in clinical and surgical wards, there were patients with all food routes. Of the 727 patients, 78 (10.7%) patients were in enteral nutrition therapy, and 23 (3.2%) patients in parenteral nutrition therapy. Oral nutrition supplementation was prescribed to 214 (29.4%) patients to complement the oral route. Finally, the 412 (56.7%) were of patients had not a prescription of nutrition therapy, remaining only with the oral route. Since this study considered quality indicators regarding enteral and parenteral nutrition, we did not show the frequency of the other routes.

Other studies also showed similar frequencies of the enteral and parenteral nutritional therapies. The Brazilian Hospital Nutritional Assessment Survey (IBRANUTRI) included 4.000 hospitalized patients in all wards, not just clinical and surgical ones. The enteral nutrition therapy was found , in 6.1% of patients and the parenteral nutrition therapy in 1.2% of them. Multicenter ELAN study conducted in Latin America involving 9.348 hospitalized patients also found 8.8% of patients (6.3% enteral nutrition and 2.5% parenteral nutrition). Thus, the frequency of patients undergoing nutritional therapy identified in the present study is similar with the others in the literature.

References

Waitzberg DL, Caiaffa WT, Correia MITD. Hospital malnutrition: The Brazilian national survey (IBRANUTRI): A study of 4000 patients. Nutrition 2001;17:573–80. https://doi.org/https://doi.org/10.1016/S0899-9007(01)00573-1.

Correia MIT., Campos ACL. Prevalence of hospital malnutrition in Latin America: the

multicenter ELAN study. Nutrition 2003;19:823–5. https://doi.org/10.1016/s0899-

9007(03)00168-0.

  1. Figure 3 is a bit confusing to comprehend - would suggest clearly spelling out the aim of each of the two bars in each category. In some categories, we want the indicator to be

less and in other, we want it to be more - this does not come across very easily until I read it a few times.

Comment to Reviewer#2

We appreciate and agree with the Reviewer. Firstly, we ordered the quality indicators for a best understanding. The first five indicators are extremely important for the good clinical and nutritional monitoring of the patient, so the health team hope to achieve and/or exceed the recommended goal. The last seven indicators, on the other hand, are adverse events in the clinical management of the patient. Because of this, these quality indicators must be avoided, so the objective of the health team is to reach values lower than the goal. After that, we try to clarify these points also in the text of manuscript, emphasizing the light bars (goals) and the dark ones (the results obtained in the study) in the text and in the legend of Figure 3. For a better understanding, please see the new sentences in the Results sections (Page 5, lines 153-158) and Figure 3 (page 6).

"…(Figure 3 -– the light bars indicate the recommended goals and dark bars indicate the results obtained in the present study.) It is important mentioning that the first five indicators are extremely important for the good clinical and nutritional monitoring of patients, so we hope to reach and/or exceed the recommended goal. The last seven indicators, on the other hand, are adverse events in the clinical management of patients that must be avoided, so that the objective is to achieve values below the goals. "

  1. Many of the references are non-English and may need either English translations or citations to English language articles based on journal policy.

Comments to reviewer

We agree with the reviewer and add citations to the English language in the references based on journal policy. Please, see the inclusion in the References section.
